# Prioritization Based Task Offloading in UAV-Assisted Edge Networks [note 1]

**DOI:** 10.3390/s23052375

**Published:** 2023-02-21

**Authors:** Onur Kalinagac, Gürkan Gür, Fatih Alagöz

**Affiliations:** 1Department of Computer Engineering, Bogazici University, 34342 Istanbul, Turkey; 2Institute of Applied Information Technology (InIT), Zurich University of Applied Sciences (ZHAW), 8401 Winterthur, Switzerland

**Keywords:** UAV networks, edge computing, task offloading, SDN, emergency computing, vehicular communications

## Abstract

Under demanding operational conditions such as traffic surges, coverage issues, and low latency requirements, terrestrial networks may become inadequate to provide the expected service levels to users and applications. Moreover, when natural disasters or physical calamities occur, the existing network infrastructure may collapse, leading to formidable challenges for emergency communications in the area served. In order to provide wireless connectivity as well as facilitate a capacity boost under transient high service load situations, a substitute or auxiliary fast-deployable network is needed. Unmanned Aerial Vehicle (UAV) networks are well suited for such needs thanks to their high mobility and flexibility. In this work, we consider an edge network consisting of UAVs equipped with wireless access points. These software-defined network nodes serve a latency-sensitive workload of mobile users in an edge-to-cloud continuum setting. We investigate prioritization-based task offloading to support prioritized services in this on-demand aerial network. To serve this end, we construct an offloading management optimization model to minimize the overall penalty due to priority-weighted delay against task deadlines. Since the defined assignment problem is NP-hard, we also propose three heuristic algorithms as well as a branch and bound style quasi-optimal task offloading algorithm and investigate how the system performs under different operating conditions by conducting simulation-based experiments. Moreover, we made an open-source contribution to Mininet-WiFi to have independent Wi-Fi mediums, which were compulsory for simultaneous packet transfers on different Wi-Fi mediums.

## 1. Introduction

Future network infrastructure such as 6G networks is expected to provide much better QoS and user experience for a wide range of networked services anytime–anywhere in an edge-to-cloud continuum [1]. To this end, Unmanned Aerial Vehicles (UAVs) are instrumental to deploy a local network infrastructure dynamically or on an on-demand basis. These circumstances can occur during short-term demand surges for capacity, post-disaster communications, or dynamic data collection [2]. In that regard, UAV networks have three salient characteristics [3]. The first one is related to their location: since they are aerial systems, they enjoy a higher probability of line-of-sight (LoS) links for connecting ground nodes and other UAVs in comparison to terrestrial systems. Secondly, UAVs can be dynamically deployed in response to emerging capacity and connectivity requirements unlike the stationary ground infrastructure. Finally, a swarm of UAVs can construct scalable multi-UAV networks in a flexible configuration for pervasive and seamless services.

On the ground level, novel intelligent mobility solutions such as self-driving cars are already in operation, and vehicle communications are becoming more vital as user adoption increases. Applicable use cases in 5G and future networks may also be mission-critical, relying on connected services, such as emergency communications or cloud robotics for search and rescue. For these applications, UAVs can assist user devices and vehicles by providing cloud access, packet relaying or edge services for computation, storage and networking. In this work, we consider such a UAV integrated edge network and focus on the use case in which UAVs create an ad hoc network for connectivity including emergency communications. The investigated edge network is software-defined as envisaged in current 5G as well as future networks, which allows our system to cope with network management issues such as mobility-based frequent topology change and dynamic system states [4]. A post-disaster operations management scenario is designed in which the network offers Vehicle-to-Vehicle (V2V) and Vehicle-to-Cloud (V2C) task offloading services to users, e.g., search and rescue team vehicles or related IoT devices.

For such use cases where we have time-sensitive and mission critical services, it is imperative to minimize delays via appropriate resource allocation while satisfying resource constraints. In this paper, we work on this problem and develop a task offloading scheme comprising different heuristics. We adopt a priority based approach for computation tasks in our edge UAV network with limited connectivity and resource availability. We also integrate a processing deadline into our scheme for each task as the key QoS criterion. Moreover, we investigate the performance of this system under different operating conditions via simulation-based experiments.

The contributions made in this paper can be listed as follows:*Offloading management optimization model:* We proposed a mathematical model to make centralized assignment/offloading decisions for the incoming task processing requests. Our model considers centralized assignment decisions for task offloading requests received from mobile ground user units. Offloading can be assigned to ground vehicles or a cloud processor service. The model’s objective function is to minimize average penalty scores caused by task completion time;*Open-source contribution to Mininet-WiFi:* The wmediumd application, a Wi-Fi delay and loss simulator and used by a Mininet-WiFi emulator [5], did not support independent mediums simultaneously, which meant that all packets experience scheduling collisions with one another. It had prevented us from performing multiple data transfers at the same time without suffering a significant loss in throughput. We have completed the development of a source-code contribution for both Mininet-WiFi and wmediumd to isolate networks automatically with a custom configuration option;*Heuristics and quasi-optimal algorithm for task offloading:* Three heuristic algorithms and a branch and bound style quasi-optimal task offloading algorithm are implemented. The latter works on the previously simulated data to obtain better insight for the evaluation of our heuristic algorithms with respect to near-optimal operation.

The rest of the paper is organized as follows: In Section 2, we provide an overview of related studies in UAV networks, task offloading, and SDN. In Section 3, we present the system-related models. Next, we develop our problem formulation in Section 4. Then, we explain proposed task offloading algorithms in Section 5. In Section 6, we describe our simulation environment and its components. Later on, simulation results and performance metrics are presented in Section 7. Finally, we conclude the paper in Section 8.

## 2. Related Work

The related work is divided into the three main categories. The first category includes works on UAV applications in vehicular networks. The second category discusses task offloading algorithms and edge-cloud continuum perspective based on UAVs. Finally, SDN use cases for UAV and vehicular networks are reviewed in the final category.

### 2.1. UAV-Aided Vehicular Networks

Vehicles communicate with each other and with public networks through various communication systems, such as cellular networks and VANETs, to improve road efficiency and safety and access connected services [6]. UAVs mounted with communication equipment can improve the connectivity and efficiency of these networks in various ways [7]. UAVs can optimize the VANET routing process or they can be utilized as Store-Carry and Forward nodes to help ground vehicular networks improve connection and reliability in the presence of disconnected vehicles.

Similar to our study, Jia and Zhang [8] use UAVs as flying base stations to provide communications to rescue vehicles in disaster-affected areas. They look into the relationship between UAV altitude and vehicle connectivity for a single UAV scenario. In the case of many UAVs, they investigate the smallest number of UAVs required to achieve a certain level of vehicle communication. In  [3], a case study on UAV-aided Vehicular Network (VN) architecture is presented, in which vehicles drive along a bi-directional two-lane straight highway while two mobile drones are flying over them to form a relay platform. They compared UAV-aided VN’s throughput and latency performance to an 802.11p-based vehicular network, demonstrating its effectiveness. These two studies focus on infrastructure level rather than possible applications such as task offloading on the constructed network.

Pourbaba et al. [9] proposed a sub-optimal method to increase coverage of full-duplex relay networks among ground vehicles by positioning UAVs. They have used a set of predefined UAV locations and have an estimated Signal-to-Interference-plus-Noise Ratio (SINR) between ground vehicles and UAVs. They have formulated an $l_{0}$-norm non-combinatorial and NP-hard minimization problem and used an $l_{1}$-norm approximation for it. With their approximation, a ten percent performance increase has been observed compared to a baseline scenario where UAVs have fixed locations.

In [10], data dissemination in UAV-aided VANETs is investigated by formulating a network throughput maximization problem to find delivery strategies and select the optimal paths for data delivery while considering the transmission rate of links and the delay constraint for data dissemination. In their analysis, the graph knapsack problem is reduced to the throughput maximization problem, and a polynomial-time approximation scheme is proposed to solve that graph knapsack problem. In another study on UAV-aided data dissemination, Zhang et al. [11] propose a protocol to reduce delay and improve system throughput by using UAVs as relayers with caching capabilities in VANETs. UAV trajectories are planned based on vehicle distributions and topology, which directly impacts the performance of UAV-assisted data dissemination. A centralized UAV trajectory algorithm (CTS-DP) is developed based on dynamic programming. Their results imply that CTS-DP outperforms the distributed trajectory scheduling algorithm (MHC) studied in their previous work. In the prior algorithm MHC, each UAV schedules its trajectory by maximizing vehicle coverage.

### 2.2. Task Offloading in UAV Networks

UAVs face challenges acting as edge computing nodes because of their computing power, weight, and battery limits. As a result, task offloading is one of the key approaches to facilitate the limited resources in the overall system as much as possible and was recently researched as part of the edge-cloud continuum paradigm.

Wang et al. [12] propose a post-disaster rescue computation task offloading scheme for cooperative UAVs. UAV computation tasks are offloaded to Unmanned Ground Vehicles (UGVs) that have idle computation resources. Both UAVs and UGVs seek their own maximum profits in its setting. They propose a stable matching algorithm to transform the computation task offloading problem into a two-sided matching problem, taking into account that the algorithm iteratively solves the problem while maximizing the utility of UAVs.

In [13], a delay-optimal task offloading approach is proposed for a multi-tier edge-cloud computing system in a multi-user environment. The main objective is the minimization of the total service time for UAVs applications. In this approach, UAV task execution is distributed across edge and cloud server computing nodes. Similarly, in [14], a dynamic heuristic algorithm is proposed to offload periodic tasks with the objectives of minimizing total time delay and energy consumption for the Software Defined Vehicular Network-supported services in the UAV-enabled MEC system.

Huang et al. [15] work on delay-sensitive task offloading to fog networks. They propose a heuristic particle swarm optimization algorithm based on a Lyapunov framework to complete tasks on time with the lowest energy consumption. In their stated scenario, computing units can be sped up to meet the time and stability constraints.

In a multi-UAV-assisted road traffic scenario, authors in [16] construct a three-player sequential game-based computational offloading methodology for processing UAV data. The computation delay, energy overhead, and communication and computation costs are all part of the utility function being studied. Contrarily, task owners in our model are mobile ground units and task offloading decisions are centralized while UAVs only provide the network.

In [17], an SDN-enabled UAV-assisted vehicular computation offloading cost optimization framework is defined. Vehicles can offload computationally intensive and time-sensitive tasks to reduce execution time and overall energy usage. Each vehicle receives network traffic data from the SDN controller. Vehicles make decentralized offloading decisions based on their own interests and global data. The stated problem is very similar to the one we addressed in our work, but task offloading to other mobile ground vehicles and UAV mesh networks is not possible in the given study contrary to ours.

As an example of multistage optimization works, a UAV-assisted mobile offloading and the trajectory optimization problem is studied in [18]. The stated problem turns out to be non-convex, and the minimum user utility is maximized under the partial offloading schemes. Their three-stage optimization algorithm is based on successive convex approximation and nonlinear fractional programming. The power allocation, bandwidth allocation, offloading decision, and UAV trajectory are jointly optimized for the solution.

In [19], Kang et al. propose a hierarchical aerial computing system that would allow high altitude platforms (HAPs) and UAVs to work together to provide computing services for ground devices (GDs) with a variety of QoS requirements. In a hierarchical aerial computing system, they use the combined computing of UAVs and HAPs to meet the QoS requirements of GDs. They develop a partially observable Markov decision method to solve the very complex nonconvex optimization problem, which is constrained by multi-dimension resource management, energy constraints, and collision avoidance.

In [20], Sacco et al. propose a self-learning strategy that assists a UAV with deciding whether to delegate its tasks. The decisions are made based on the agent’s projected behavior, which suggests whether or not edge cloud is advantageous to the incoming tasks. To make a prediction about future device load, they either use a model from the time-series class or a model from the class of ML regressors.

Zaman et al. [21] propose a task offloading framework with light user mobility prediction utilizing machine learning techniques. Based on past data, they predict the user’s future location and may decide on the task offloading at an edge server which is close to the location instead of local task processing. They compare their task offloading algorithm with two other mobility based algorithms from the literature.

### 2.3. Software-Defined UAV Networks

SDN allows for network programmability by separating the control layer from data-forwarding network elements. SDN architecture can be used to reduce network configuration time and cost, which is critical for rapidly changing UAV networks [22]. It can also handle offloading requests and associated routing management based on network traffic [23].

Privacy and protection against external cyber threats are both possible by exploiting the capabilities of SDN architecture. In that regard, important application domains are analyzed from a security standpoint in [24]. The description and classification of performance assessment metrics in drone design are discussed for test-bed based approaches, most widely used simulation platforms, and hybrid methods. In addition, potential vulnerabilities for drone-enabled SDN-based societies are presented.

An SDN routing framework for UAV networks is developed in [25]. A monitoring platform to obtain network statistics for SDN controllers and a load balancing algorithm based on that platform’s analytical results are introduced. Routing decisions are controlled by considering the power limit of UAVs and the priority of flows.

For UAV-assisted infrastructure-less vehicular networks, Alioua et al. [26] propose a distributed SDN-based architecture. They also propose a road safety scenario as a use-case for their design. UAVs are used in this case to help emergency rescue vehicles in exploring and investigating inaccessible affected zones. Lastly, they formulate an offloading decision problem based on a two-player sequential game and prove the existence of Nash equilibrium.

Zhu et al. [27] consider an SDN-based cellular network with a UAV and BSs. UAV serves as a computational server in the system, calculating users’ tasks, or as a relay node, forwarding users’ jobs to MEC-enabled BSs, where they can be calculated in MEC servers. The objective function is formulated as minimization of the total weighted delay and energy consumption of the UAV and all users. They propose a joint mode selection and resource allocation optimization algorithm to solve this optimization problem. The proposed algorithm decomposes the optimization problem into two subproblems: task mode selection and resource allocation, which are performed in alternating iterations.

## 3. System Model

Our network model consists of $N_{U}$ UAVs ($U_{i}$) and one ground base station (BS). UAVs are mounted with wireless Access Points (APs) having multiple Wi-Fi interfaces. One of the interfaces for each AP provides Wi-Fi service to terrestrial vehicles while the rest form mesh networks with nearby UAVs’ APs. One of the UAVs is connected to the BS which has a backhaul link to the Internet and relevant cloud services. A computational unit that houses an SDN controller and serves as the network controller is also mounted on one of the UAVs. This controller is managed by a service provider remotely. UAVs are primarily used for communication, which reduces energy consumption, rather than to offload tasks. We do not take energy constraints into account for the sake of simplicity. The notations used in system modeling are listed in Table 1. The network model is shown in Figure 1.

All the task offloading decisions are centrally made by a controller application, namely the offloading orchestrator. Two main types of terrestrial vehicles are present in the system. The first type corresponds to rescue and emergency vehicles, which are task owners and signal their task information including a deadline, priority factor, and size to the task pool. This pool is located at the offloading orchestrator on the controller. The orchestrator application assigns these tasks to the second type (compute station) of vehicles, also called task processors, and informs both parties about this assignment. Only task identifier data are essentially stored in the task pool, not the task data itself. Therefore, the data transfers are carried out directly from the task owner to the processor units. In Figure 2, this basic task assignment flow is given. The processor vehicles have their own limited-size queues to store unprocessed tasks. A processor may simultaneously download the data of multiple tasks if the policy of the active task offloading algorithm allows that. The processing speed and queue size depend on the processor group (i.e., we assume class-based processor capabilities). Similarly, the priority of a task is determined by the owner vehicle’s group (i.e., class-based task priority). If a task is not completed by the deadline, the system receives a penalty proportional to the amount of time that has passed between the deadline and the task completion time. The mobility of vehicles is not linked to the task processing operation, i.e., a vehicle may leave the area, regardless of its tasks still being processed. The tasks are preserved for vehicle departures. In other words, a task is also sent back to the pool if the assigned processor vehicle leaves during packet transmission for that task. Depending on the deadline, two cases may occur for the situation in which a task owner leaves the coverage area: The system receives an extra penalty score *z* if the deadline is already missed; otherwise, it receives no penalty. To identify and monitor the network location of vehicles, the orchestrator application listens for ARP packets. Additionally, access points notify the controller of vehicle departure or disconnection events at their associated stations periodically.

The tasks created by the vehicles have an exponentially distributed inter-arrival time with rate λ. If the task pool is not empty when a new task arrives, it may be offloaded/migrated to the cloud processing server with a probability based on the task pool’s current occupancy at that time. Essentially, we consider minimizing average prioritization-based weighted penalty points for task offloading management as a QoS key performance indicator and thus propose algorithms to achieve that objective.

### 3.1. Channel Models

Since there is no consensus on a proper path loss model for UAV-enabled networks [28], we selected two popular path loss models [29,30] and applied them in our network simulations. In our simulation environment, the data rates of connections are probabilistic, and the probability values are obtained from the Received Signal Strength Indicator (RSSI) transmission mode matrix. The matrix values contain the probability of the Wi-Fi rate at the given RSSI value, i.e., 6, 9, 12, 18, 24, 36, 48 and 54 Mbps for the IEEE 802.11g standard. The RSSI values for each connection are estimated based on the utilized channel models. The channel model equations used in our system are given in Appendix A.

### 3.2. Mobility Model

UAVs are deployed to particular locations to cover the area uniformly. They have small-scale mobility where these aerial systems are assumed to hover around these points in a limited space. Specifically, they are assumed to be forming a small-sized 8 shape by moving around two fixed points at the same altitude as the mobility model in [31].

## 4. Problem Formulation

The problem that we address is the minimization of the average task penalty weighted by task priority in a software-defined edge network with integrated UAVs. Due to departures, some tasks may not be fulfilled even in the optimal scenario which prevents us from using a deadline as a constraint so we introduce a penalty scoring system in which an extra penalty score is added for unfilled tasks in order to encourage the completion of all tasks. The list of time-related notations that we utilize in the problem formulation is given in Table 2. In addition, we utilize indicator function $1_{w}$ that is assigned to one if *w* is true, zero, otherwise.

A task’s lifetime in our edge network is composed of various components due to pool wait, transmission time, processor queue wait and processing time. $C_{i}$ variable represents if $i^{th}$ task $K_{i}$ is successfully processed. Parameter $x_{ij}$ is equal to one if $K_{i}$ is decided to be offloaded to $P_{j}$ and, similarly, $y_{i}$ is equal to one if $K_{i}$ is decided to be offloaded to the cloud. These decision variables are equal to zero otherwise. $t_{total,i}$ is the total time needed to complete $K_{i}$ and defined as follows if and only if (↔) the task is completed, which is the case when $C_{i}$ equals 1: (1)$$\begin{array}{c} C_{i}=\sum_{j=1}^{N_{P}}x_{ij}+y_{i}\;\;\;\;\;\;\;\;\;\;\;\forall i \end{array}$$(2)$$\begin{array}{cc} t_{total,i}=t_{pool,i}+t_{tx,i}+t_{queue,i}+t_{process,i} & \;↔\;C_{i}=1 \end{array}$$(3)$$\begin{array}{cc} t_{process,i}=S_{K_{i}}/\left(\sum_{j=1}^{N_{P}}\beta_{i}x_{ij}+\beta_{cloud}y_{i}\right) & \;↔\;C_{i}=1 \end{array}$$(4)$$\begin{array}{cc} t_{tx,i}=S_{K_{i}}/\left(\sum_{j=1}^{N_{P}}L_{K_{i},P_{j}}\left(t\right)x_{ij}+L_{K_{i},cloud}\left(t\right)y_{i}\right) & \;↔\;C_{i}=1 \end{array}$$
Equation (Equation 1) ensures that, in the best scenario, a task is processed only by a single processor. In Equation (Equation 2), the required time to complete $K_{i}$ is calculated by summing all the related time components. The related components, processing time $t_{process,i}$ and data transmission time $t_{tx,i}$, are calculated in Equations (Equation 3) and (Equation 4).

$D_{i}$ is the penalty for $K_{i}$ and calculated as:(5)$$\begin{array}{c} D_{i}=\left\{\begin{array}{c} max\left(t_{total,i}-t_{deadline,i},0\right)\;\mathrm{if}\;C_{i}=1\\ 0\;\mathrm{if}\;\left(C_{i}=0\right)\wedge \left(O_{i,V_{j}}=1\right)\wedge \left(b_{V_{j}}<a_{K_{i}}+t_{deadline,i}\right)\\ b_{V_{j}}-a_{K_{i}}-t_{deadline,i}+z\\ \;\;\;\mathrm{if}\;\left(C_{i}=0\right)\wedge \left(O_{i,V_{j}}=1\right)\wedge \left(b_{V_{j}}>a_{K_{i}}+t_{deadline,i}\right) \end{array}\right. \end{array}$$

The weighted penalty point of $K_{i}$ is calculated as:(6)$$\begin{array}{cc} w_{i}= & \alpha_{i}\cdot D_{i}\;\;\;\;\;\;\;\;\;\;\forall i \end{array}$$
where $\alpha_{i}$ is the task priority.

Now, let us present the optimization problem, which will be solved by the orchestrator application to decide on the task offloading at each time slot as follows:(7)$$\begin{array}{cc} \underset{x,y}{min}\;\; & D\left(\lambda ,R\right)=\sum_{i=1}^{N_{K}}w_{i}/N_{K}\\ \; & \mathrm{subject}\mathrm{to}: \end{array}$$(8)$$\begin{array}{cc} \; & \sum_{j=1}^{N_{P}}x_{ij}+y_{i}\leq 1\;\;\;\;\;\;\;\;\;\;\;\;\;\forall i \end{array}$$(9)$$\begin{array}{cc} \; & x_{ij}\cdot c_{K_{i}}\leq b_{P_{j}}\;\;\;\;\;\;\;\;\;\;\;\;\;\;\;\;\forall i \end{array}$$(10)$$\begin{array}{cc} \; & x_{ij}\in \left\{0,1\right\}\;\;\;\;\;\;\;\;\;\;\;\;\;\;\;\;\;\;\;\forall i,\forall j \end{array}$$(11)$$\begin{array}{cc} \; & y_{i}\in \left\{0,1\right\}\;\;\;\;\;\;\;\;\;\;\;\;\;\;\;\;\;\;\;\;\forall i \end{array}$$
In Equation (Equation 7), we minimize the summation of the weighted penalty points of all tasks. Equation (Equation 8) ensures that only one vehicle processor or cloud is assigned to a task. There is inequality because some tasks may not be completed due to the randomness of processor availability and task owner departure times. The task completion time slot, $c_{K_{i}}$ on the left-hand side of Equation (Equation 9) is the sum of the task arrival time, $a_{K_{i}}$ and $t_{total,i}$. Equation (Equation 9) guarantees that the task should be completed before the processor departs. Because the departure times are not known during the run-time, only our quasi-optimal algorithm can use this constraint to guide decisions. For heuristics, a task is sent back to the pool if the constraint happens to fail. Equations (Equation 10) and (Equation 11) represent the possible values of the decision variables.

Task offloading decisions significantly affect multiple delay results, especially when related vehicles share the same APs or links for connectivity since available data rates for those vehicles drop. Thus, each possible decision combination should be evaluated. This multi-user, multi-task problem is proved to be NP-hard [32], while the penalty scoring in our system model increases the complexity due to its nonlinear behavior upon task owner departures.

## 5. Task Offloading Algorithms in Our Model

In our algorithm design, the system time is discretized into steps to perform offloading decisions on the generated time steps. Obviously, the exact time a vehicle will arrive/depart or a task arrives is unknown during the run-time. Performance comparisons are made using four different algorithms. The first three algorithms are greedy heuristics and are used during the simulation experiments. They make decisions according to the available data at each step, i.e., run in an online mode. They are namely *Aggressive-Wait*, *Aggressive Tx-Order*, and *Adaptive* offloading algorithms. The available data include vehicle-UAV associations, existing links with their bandwidths, and rough vehicle departure times. The final algorithm is named the *Quasi-Optimal* offloading algorithm and an oracle-type decision-maker since it operates by extracting better solutions from recorded past data for benchmarking. Since this algorithm cannot process all the possible solutions from all the data recorded due to combinatorial explosion, it is quasi-optimal by processing events in time windows. The notations used in these algorithms are listed in Table 3.

### 5.1. Aggressive Wait Offloading Algorithm (AGG-1)

The orchestrator application with the first greedy algorithm AGG-1 tries to offload tasks until all the queues on the available processors are full. New assignments to the same processor can be made before the current assigned task’s data transfer is completed. If the next assigned task is downloaded before the first one, the second task will be processed after the first is completed. The rationale is to complete the previously assigned task, probably being the one closer to its deadline, earlier. The AGG-1 algorithm is described in Algorithm 1, and it is run until the simulation time is up.
**Algorithm 1:** Aggressive-Wait Offloading Algorithm (AGG-1) 
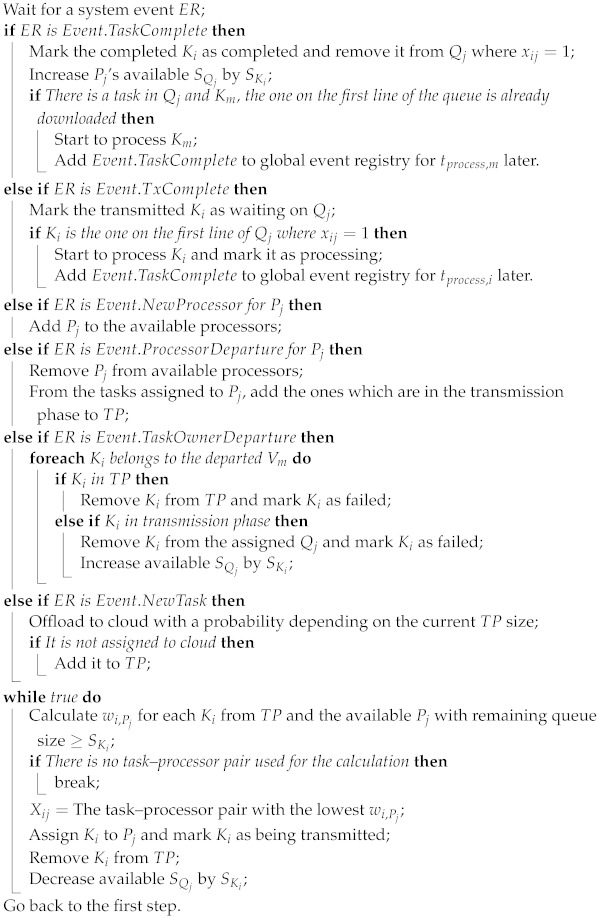


### 5.2. Aggressive Tx-Order Offloading Algorithm (AGG-2)

The second greedy algorithm is similar to the first one, AGG-1. The only difference is that the task transmission completion order (Tx-Order), rather than the task assignment order as it was in AGG-1, determines the order in which tasks are processed; the one downloaded earlier is processed first. This difference corresponds to the handling of Event.TaskComplete and Event.TxComplete events in Table 1, and the related processor does not check whether the downloaded task was the first one assigned before processing, contrary to AGG-1.

### 5.3. Adaptive Offloading Algorithm (ADP)

In ADP, during an offloading decision, the processors with the tasks being transmitted are skipped, avoiding multiple transfers for the same processor. It aims to decrease the pressure on the limited network resources and to improve prioritization since it leads to more tasks in the controller task pool TP and more processor-task combinations upon arrival of new processor nodes. Algorithm 2 shows the details of the algorithm.
**Algorithm 2:** Adaptive Offloading Algorithm (ADP) 
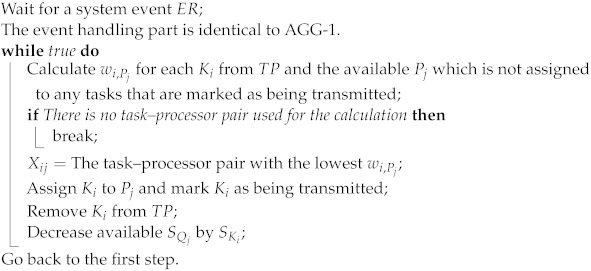


### 5.4. Quasi-Optimal Offloading Algorithm (Q-OPT)

The last approach is a branch and bound based algorithm that processes the whole simulation data to produce a sub-optimal yet close to optimal objective value for benchmarking our heuristic algorithms. It analyzes the data from past simulations and functions as an oracle-type decision-maker. This processed data includes task details (e.g., arrival time, size, deadline), vehicle positions, vehicle associated AP and RSSI values, and task processor details.

For each task, possible processor assignments are determined as lists. In order to narrow down the candidate solutions, arrival and departure times of vehicles and their connection status per step are considered to filter out the unfitting processors on each list. Two more decisions, skipping (meaning failing on purpose) and offloading to the cloud, are also added to these lists. Even the most underloaded setup with meaningful results generates a huge candidate pool from the Cartesian product of decision possibilities, which cannot be handled due to combinatorial explosion. Thus, we have developed a time-windowed quasi-optimal processing approach. The solver processes and makes decisions on the tasks in the given time interval. The algorithm starts with the time window to check if the candidate solution pool can be processed. As illustrated in Figure 3, it shrinks the window end time (window length) until the number of decision combinations is less than a predefined threshold, $\Theta_{decision}$. The solver skips some of the combinations by bounding. After finding the optimal value for this interval, it saves the decision of the first task in this window and moves to the second time window, which is between the arrival time of the next task and the end of the simulation. It applies the same principle until all tasks are decided one by one.

## 6. System Implementation

For our tests, we use an Ubuntu 21.04 operating system on a mobile workstation with an Intel Core i7-10875H (2.30 GHz × 8), 16 GB RAM, and 256 GB SSD storage. The docker version of ONOS [33] v2.8 is utilized as an SDN controller. Mininet-WiFi [5] is used to create host devices, virtual stations, and APs. On Mininet-WiFi and its wireless medium simulation tool dependency wmediumd, some crucial improvements had to be made in order to support independent and simultaneous Wi-Fi packet transfers. They are merged into the main branch (https://github.com/intrig-unicamp/mininet-wifi/pull/419, accessed on 19 December 2022, https://github.com/ramonfontes/wmediumd/pull/9, accessed on 19 December 2022). Open vSwitchv2.15 [34] is used for virtual switch emulation. For traffic generation from vehicles in our experiments, iPerf v3.10 [35] is used. Eclipse SUMO v1.11.0 [36] is used for vehicle mobility simulation. We implemented our main simulation driver application in Python 3.9 and the offloading orchestrator in Java 11. The baseline simulation parameters are listed in Table 4.

### 6.1. System Components

There are four main application modules for implementing and running simulations for our task offloading system in a UAV-assisted software-defined edge network environment. These are *Vehicle Mobility Generation*, *Network Infrastructure*, *Offloading Orchestrator*, and *Main Simulation Driver*. The overall system is depicted in Figure 4.

#### 6.1.1. Vehicle Mobility Generation

We chose a one-square-kilometer area from Istanbul’s Besiktas District (Figure 5) to reflect an urban traffic and road infrastructure in our system. The data from the selected region is imported by using the OSM Web Wizard, a utility application bundled with SUMO. On each run, we retrieved location data from the SUMO Traffic Control Interface (TraCI) iteratively while tracking the progress on the GUI interface. To visualize the topology better, UAVs and the BS are inserted as point-of-interest objects on the GUI interface and updated at each step. Additionally, circles are used to highlight APs’ terrestrial ranges on the interface. In our simulations, we used five different vehicle classes: two of them with the task owner role, two of them with the task processor role, and one with no network role. The last one represents private vehicles that our network does not serve. Vehicle related simulation settings are listed in Table 5.

#### 6.1.2. Network Infrastructure

The mac80211_hwsim software is a Linux kernel module, and it can virtualize Wi-Fi network cards. Mininet-WiFi, a fork of Mininet [37], initially loads this module with the desired number of virtual wireless interfaces necessary for our system. These interfaces can be configured just like the real ones, and Mininet-WiFi automates this process. mac80211_hwsim tracks the current channel of each card’s radio and copies all transmitted frames to all the operating radio within the same channel. Mininet-WiFi also benefits from wmediumd, an application which allows simulating frame loss and delay by managing mac80211_hwsim from its netlink API. It has an event mechanism and adds artificial delays and failures to packet forwarding operations depending on the given channel model and radio locations. Dynamic configuration on these parameters can be performed via its socket API.

All Wi-Fi networks are contention-based and time-division duplexing systems [38]. In wmediumd, a queue mechanism is introduced to order packets to reflect delay caused by this concept. However, when multi-data transfers are performed simultaneously, it causes undesired low throughput levels even for the networks that are separated by distance or channel. There is a study on this issue that suggests a multi-threaded architecture from scratch [39]. To enable our experiments, we have developed a multi-medium mechanism on top of the existing repositories, making it possible to isolate queues of the nodes in different mediums. The new version of the application detects which nodes are talking to each other by sniffing AP-Station packet transmissions and grouping them. The code improvement also gives manual configuration ability to Mininet-WiFi end users to setup the system for advanced topologies. In our infrastructure, the average CPU usage by wmediumd during simulation was no more than 25% both before and after the multi-medium development, which indicates that our extension did not cause additional overhead and there was no CPU bottleneck in both cases.

A LoS probability-based channel model is also implemented on both Mininet-WiFi and wmediumd for AP-Station interfaces while keeping the existing log distance model for mesh interfaces. Thus, the applications are run in dual channel model mode. Unfortunately, a public contribution has not yet been made because it is currently a work in progress.

Nine APs representing UAVs and one AP representing the BS have been added to the network. UAVs are placed in a grid formation, and one of them is randomly selected for BS connection. A maximum of three separated mesh links per AP are allowed, and the links are selected by our routing algorithm, which tries to maximize bandwidth from BS to the rest of the APs after creating a minimum spanning tree with the highest link capacities. For AP-BS and AP-AP connections, Wi-Fi 802.11s mesh protocol is used. Different AP and mesh channels are assigned to the neighbor device interfaces. For the cloud server role, a Mininet host is linked to the BS AP. For the controller host role, another host is connected to the centrally positioned AP. A *localhost* machine is also linked to the system by using the Network Address Translation (NAT) feature of Mininet for debugging purposes. The iperf3.10 tool is used for its fresh ’zero copy’ feature to lower the CPU burden caused by the app. For each data transmission, server and client apps are run on the related station and host interfaces.

#### 6.1.3. Offloading Orchestrator

Two ONOS controller applications are developed, one for traffic monitoring and one for packet routing. The first one is used for testing purposes, which collects traffic data on device interfaces. On the other hand, the second one is one of the core developments required to facilitate the network. It works as an in-built reactive forwarding app, but there are crucial differences: When a new ARP packet is received, the in-built app floods the packet while our app installs all the flow rules required for end-to-end communication. Our application also installs ARP flow rules to lead ARP packets through the desired connection path, allowing mesh nodes to store host data. There are multiple separated mesh networks within the system. Mesh connections of APs do not pass data packets if they have not discovered the host beforehand in the current mesh protocol, IEEE 802.11s [40]. If a network location change is detected, the application invalidates all the previously installed flows related to that vehicle. A running sample of our topology is shown in Figure 6.

#### 6.1.4. Main Simulation Driver

A modular Python application handles the majority of the simulation logic. The simulation app performs its routines each second and then sleeps. Upon receiving vehicle locations from SUMO TraCI, it updates the location of the stations on Mininet-WiFi. It can send direct commands to a station and host to trigger task generation signaling events and V2V task data transfers. The controller is assumed to be mounted on the central UAV, so the application triggers Netcat client command from the station associated with the task owner vehicle.

## 7. Performance Evaluation

For the evaluation of the proposed system and the efficiency of our heuristic algorithms, we have selected two main scenarios and three different cases for each. As a baseline algorithm, we included the Only-Cloud option in which all tasks are offloaded to the cloud. Each simulation run lasts for 15 min, including pre-population and cool-down periods, and emulates the same amount of time. All the cases are run ten times for each case and algorithm combination. The results are presented in the following section.

### 7.1. Scenario 1—Task Inter-Arrival Time (1/λ)

The expected time value between two tasks is exponentially distributed and equals 1/λ. This scenario shows how the system responds to different loads (varying task request intervals). The different values for the interval 1/λ are:*Case 1.1:*1/λ = 5 s;*Case 1.2:*1/λ = 10 s;*Case 1.3:*1/λ = 15 s.

### 7.2. Scenario 2—Average Vehicle Speed $\bar{V}$

Starting from the average walking speed of 5 km/h, we analyze the default case of 20 km/h and the relatively high-speed case. As the speed increases, vehicles become more likely to depart from the network coverage. Additionally, fast movement means more handover, resulting in more overhead:*Case 2.1:*$\bar{V}$ = 5 km/h;*Case 2.2:*$\bar{V}$ = 20 km/h;*Case 2.3:*$\bar{V}$ = 40 km/h.

### 7.3. Experimental Results

In this part, we give the numerical findings from our experiments and use them to analyze the performance under two different scenarios. First, we compare each algorithm’s objective function D, which represents the average of penalty points weighted by task priorities. Second, we determine their task processing characteristics by analyzing the task failure and cloud offloading ratios along with controller pool time (${\bar{t}}_{pool}$). Lastly, we analyze their temporal results for task completion by considering the averages of the task completion (${\bar{t}}_{total}$), transmission (${\bar{t}}_{tx}$), processor queue (${\bar{t}}_{queue}$), and processing (${\bar{t}}_{process}$) time values.

#### 7.3.1. Impact of Task Inter-Arrival Time

In Scenario 1, we applied three different task request inter-arrival times while maintaining the same expected number of vehicles. The APs deployed on the environment use 802.11 g mode, which has theoretical throughput of up to 54 Mbps but actually achieves much less due to overhead and path loss, so there was network congestion even at the *Case 1.3*, which has the highest task inter-arrival time. The maximum throughput in our simulation environment was 36 Mbps, which was achievable for a vehicle-to-UAV link under 150 m in length. Since many of the vehicles utilize the same interfaces, their share of the bandwidth becomes much less.

iImpact of Task Inter-Arrival Time on Penalty Results

Our first and main evaluation criterion is the performance analysis of the algorithms in terms of the main objective function D, which aims to minimize the weighted average of the prioritization-based penalty points. In Figure 7, the impact of varying task request intervals on D for each algorithm can be seen. A cloud connection is more stable than a connection between two moving vehicles because the cloud side is stationary. Cloud offloading also has the advantage of faster processing and no queue waiting time. As a result, the Only-Cloud option performs better compared to other heuristics under light load. On the other hand, as request interval decreases, all algorithms exhibit an exponential increase in D. Among them, the Only-Cloud option is the most impacted one because the backhaul link to the cloud becomes a bottleneck, resulting in slower packet transfers, whereas our heuristics can distribute load to different processors.

The main logic of AGG-1 for preserving download starting order for task processing is to wait for tasks that are older or chosen earlier due to their priority-deadline values. It does not seem to be making a meaningful difference in terms of D compared to the non-preserving version, AGG-2. Having oracle-type decisions and being theoretical output, Q-OPT seems to give superior results compared to the rest, especially as the load increases since it can make better decisions in advance while the rest make decisions with much less limited data. Among the online decision-makers, ADP outperforms the competition by a significant margin in all three cases.

Along with D, we can see the service delivery skills of the algorithms in Table 6. For Case 1.1, in which the system is in its most loaded state, users face a high ratio of task offloading failures, which is due to task owners leaving the area before their task is successfully transmitted at a high percentage. In addition, an increase in the cloud ratio means the number of tasks in the controller pools has been increased. ADP has the highest cloud offloading ratio, which is expected since the processors do not download multiple tasks simultaneously, leading the pool to be more crowded. Still, ADP has the best service scores compared to AGG-1 and AGG-2, considering D and task failure ratios. After analyzing the experimental data records, we have discovered two main reasons. Firstly, the task download ratio per processor is always less than 1, which is not the case for the other two heuristic algorithms. Thus, it leads to a less congested network and current offloading tasks’ transmission times to be shortened. Secondly, since the task pool gets filled more easily, utilization of the cloud commences earlier, while AGG algorithms start to offload to the cloud after all their queues are full.

iiImpact of Task Inter-Arrival Time on Task Processing Characteristics

From Table 6, we observe that AGGs achieve lower average pool time ${\bar{t}}_{pool}$ resulting from their aggressiveness to fill processor queues. However, ADP and Q-OPT achieve better performance results in terms of the objective function since they complete tasks one by one without getting the infrastructure congested. Another reason is that Q-OPT and ADP allow for easier management of network load by delaying decisions in order to obtain better assignment options from vehicles that are arriving in the area or from vehicles that were not previously available.

iiiImpact of Task Inter-Arrival Time on Temporal Results for Task Completion

In Figure 8, the component-based average packet delays are presented in a stacked form. The labels on the top of the bars indicate the task inter-arrival time used for the given test group. Numerical values are also represented in Table 7. By observing the task life cycle, we can obtain more insights about the system operation and the potential improvements. The first thing to notice is that transmission delays account for the majority of system delays. Increasing available bandwidth while improving network congestion management and routing protocols could be highly helpful for the performance of such a network. The processing time has little effect with the current parameters. Only on AGG-1 is a 5–10% delay increase observed due to the queue wait procedure.

The vehicles stay in the area for an average of 320 s with the current settings. By combining this data with the average system delay and task failure ratios, we can state that it is still possible to have a lower task inter-arrival time than 5 s, which would increase ${\bar{t}}_{total}$ to over 300 s, without fully congesting the network.

As a final analysis of this scenario, the system delay and task deadline of the tasks from simulations using the ADP algorithm are shown in a scatter graph in Figure 9a for Case 1.1, in Figure 9b for Case 1.2, and lastly in Figure 9c for Case 1.3. For each case, data from ten runs are reflected in the figures. It gives a good illustration of the strong effect of task generation rate on the system delay. In Case 1.1, high deviation around the average implies the system service instability at this level of load. Contrarily, in Case 1.3, our system runs at a steady state.

#### 7.3.2. Impact of Average Vehicle Speed ($\bar{V}$)

Vehicle mobility inherently has a significant effect on our framework since it changes the network topology. During handover, our offloading orchestrator reconfigures all user-related flows, but Wi-Fi reconnection and flow installation cause observable disruptions in data transfer. In the baseline case from the previous scenario, vehicles were moving at 20 km/h so, for comparison, we had chosen human walking speed as 5 km/h for low mobility and 40 km/h for the double speed comparison. In order to deal with a moderately loaded network, we set the inter-arrival task interval (1/λ) to 8 in this scenario.

iImpact of Average Vehicle Speed on Penalty Results

From Figure 10, we can see that mobility had a negative impact on performance. There are several main reasons according to the experiment data analysis. The first is that there has been more handover inference to data transfers, and the second is that offloading decision validities has lasted less; for example, task-assigned processors with good network connectivity have stayed in this mode for a much shorter duration. Only-Cloud baseline performance is the least affected by the vehicle speed changes due to its single-side stationarity. ADP seems to be the most efficient compared to the other heuristics and the Only-Cloud option.

iiImpact of Average Vehicle Speed on Task Processing Characteristics

In Table 8, we can see the service-related details. The task failure ratio for the Case 2.3 is high in general. Remembering that the task inter-arrival time is constant throughout the scenario, an increase in ${\bar{t}}_{pool}$ and cloud ratio indicates that the offloading orchestrator has issues serving incoming requests.

iiiImpact of Average Vehicle Speed on Temporal Results for Task Completion

The delay distributions are illustrated in Figure 11. As the vehicles become faster, handover overhead occurs due to the disconnections during network changes, increasing the transmission times. For all methods, the existing increase in failure ratio indicates that the offloading orchestrator has issues serving incoming requests as the speed increases.

### 7.4. Algorithm Analysis

Our algorithms compare all possible task–processor pairings, which makes them exponential. Heuristic ones consider only the currently available processors, so the solution pool is small and the problem can be solved in real-time. In Q-OPT, we limit the size of the solution pool by adjusting the considered time interval on each step in order to solve it in a feasible amount of time.

### 7.5. Discussion

To summarize, we were able to describe the core fundamentals of our proposed UAV-assisted Software-Defined Edge Network and task offloading heuristics through a step-by-step examination of the test scenarios. These scenarios focused on two main aspects: task request inter-arrival time and the average speed of the vehicles.

The first scenario helped us to investigate the capacity of the designed network under three different traffic requirements. Unfortunately, many of the vehicles share the same interfaces, which impacts their available transmission bit rates and leads to low capacity levels. From the second scenario, we observed that vehicle mobility has a big impact on our system since it dynamically modifies the network topology. During the handover, all user-related flows should be modified. Our offloading orchestrator handles this job, but reconnection and topology discovery-related delay overheads occur during topology changes.

The Only-Cloud is sufficient when the request rate is low. However, we clearly observe that MEC-based solutions are necessary in high-load cases due to the bottleneck on the back-haul connection. After investigating the performance of our algorithms, it can be said that ADP outperformed the other heuristic algorithms. This result is related to the network’s core problem: it requires effective congestion control methods to handle high levels of traffic load. If we compare AGG-1 and AGG-2 with each other, they have similar objective function results, but the average delay is lower in the more aggressive case, AGG-2, due to the extra queue time in AGG-1.

Q-OPT appears to outperform the others, especially as the service request rate increases because it can make better decisions in advance while the others make decisions with much fewer data in real-time. Q-OPT highly utilizes cloud server processing from the beginning and thus slows down when the network is congested. In addition, it fails to complete the processing of some tasks on purpose. It does not serve the task owners with their departure time earlier than their task’s deadline even when the system is idle and the task can be handled. In the high-load cases, the algorithm also skips some tasks for the sake of a better overall score.

## 8. Conclusions

From the development side, several paper related artifacts have been developed in this work. Firstly, a core application orchestrating all the components of emulation has been created. An orchestrator application for ONOS has been developed for the reactive management of flows and hosts. Open-source contributions to Mininet-WiFi and wmediumd have been made on a crucial feature that allows users to operate in isolated Wi-Fi mediums. Three heuristic algorithms and a branch-and-bound style quasi-optimal task offloading algorithm were developed, and their performance analysis along with the Only-cloud baseline case has been conducted in terms of request rate and mobility.

When the demand in the system is low, the Only-Cloud approach is generally adequate. On the other hand, we note that MEC-based solutions are especially required in high-load scenarios because of typical back-haul connection bottleneck and latency increase. Since it can make better decisions in advance while the others must make decisions with considerably less information in real-time, Q-OPT outperforms the others in terms of the objective function, particularly as the number of service requests rises. Among the heuristic algorithms, ADP achieves better performance results since the processors in ADP handle tasks one at a time without overloading the infrastructure.

The first direction for future work is to expand our framework for smart routing. The most serious issue we have encountered in our system tests is the limited capacity of wireless links. Smarter network congestion management and routing algorithms can help us to improve the capacity of the system and mitigate this issue. It could also be worth developing smarter heuristic approaches for task offloading. Lastly, better management of UAV mobility could be used to handle area-based congestion. We have observed that some of the APs are more likely to be active during our simulations. According to our analysis, this is because of the traffic model which leads vehicles, especially to the central areas in the simulated geographical area.

## Figures and Tables

**Figure 1 sensors-23-02375-f001:**
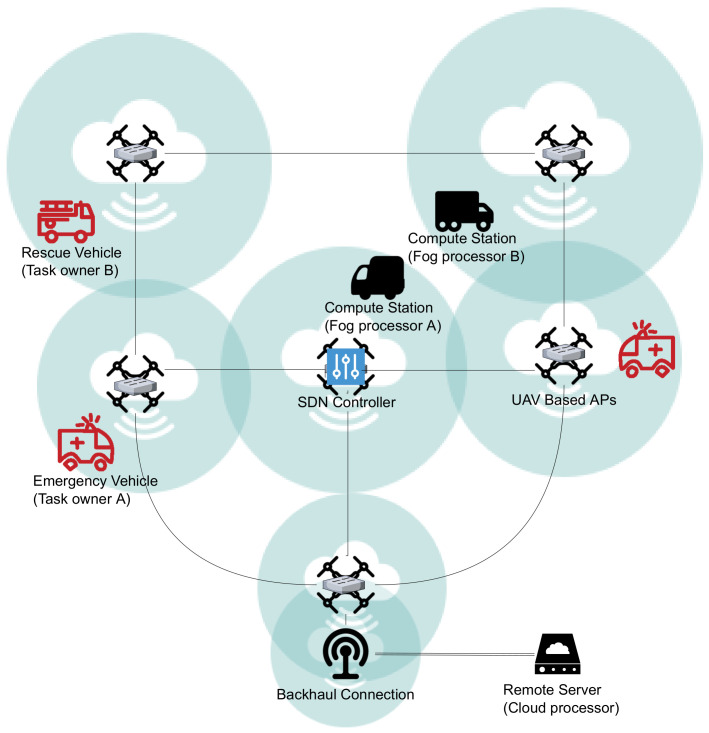
Multi-UAV edge network model investigated in our work.

**Figure 2 sensors-23-02375-f002:**
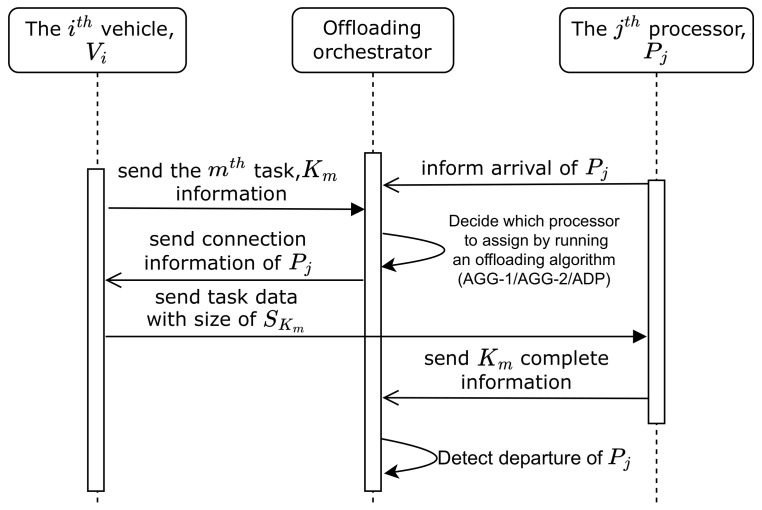
Task offloading sequence diagram.

**Figure 3 sensors-23-02375-f003:**
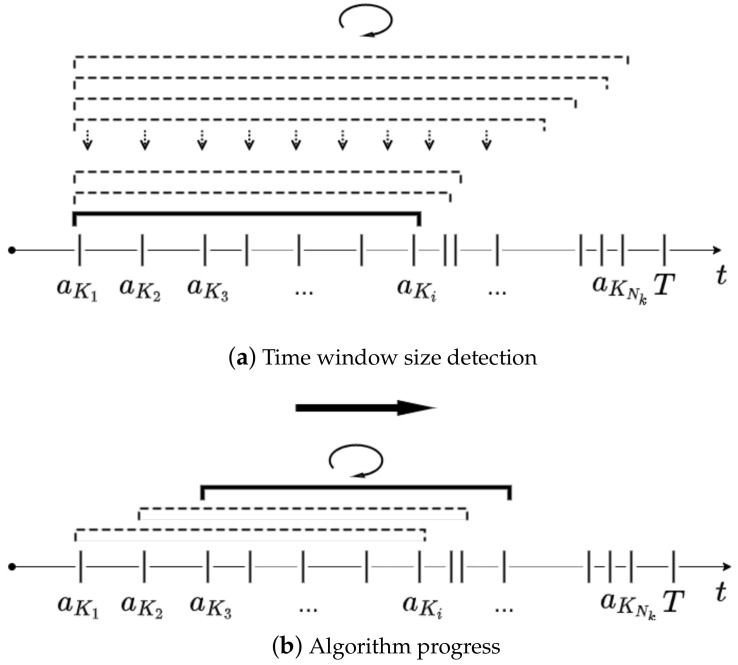
Q-OPT algorithm-time window size detection on each step.

**Figure 4 sensors-23-02375-f004:**
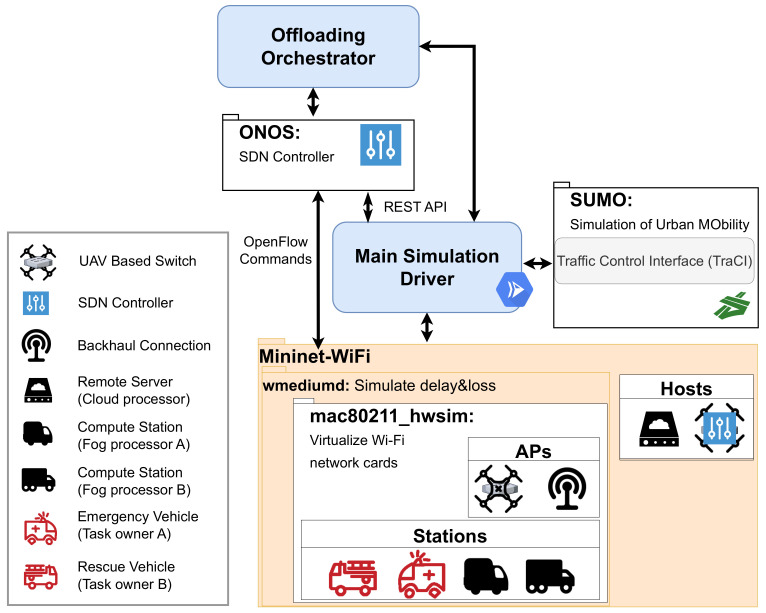
System components.

**Figure 5 sensors-23-02375-f005:**
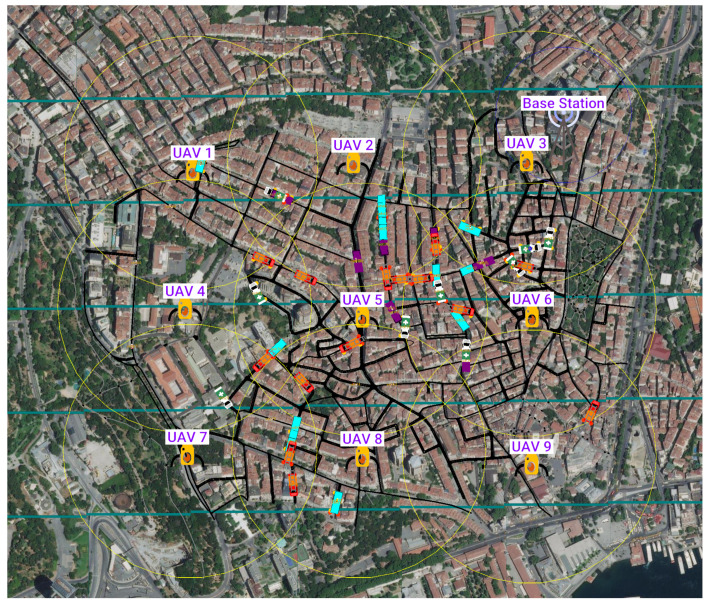
The visualization of topology elements in Besiktas network. The terrestrial coverage ranges of UAVs are indicated by the yellow circles around them.

**Figure 6 sensors-23-02375-f006:**
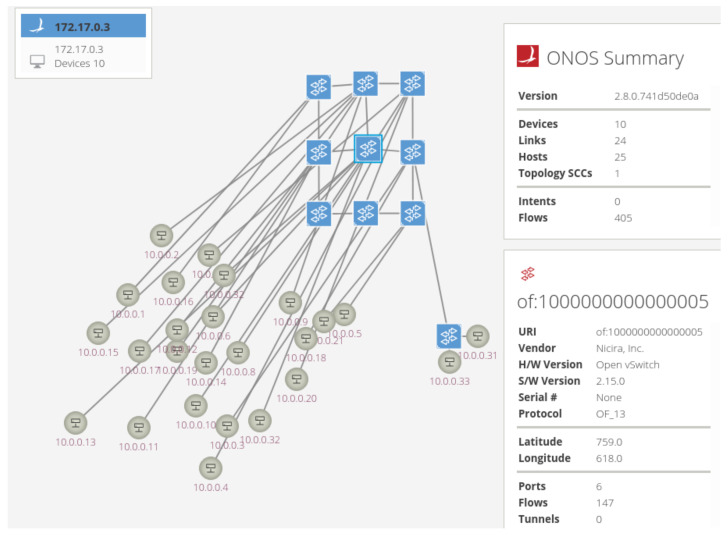
A sample network topology in the simulations.

**Figure 7 sensors-23-02375-f007:**
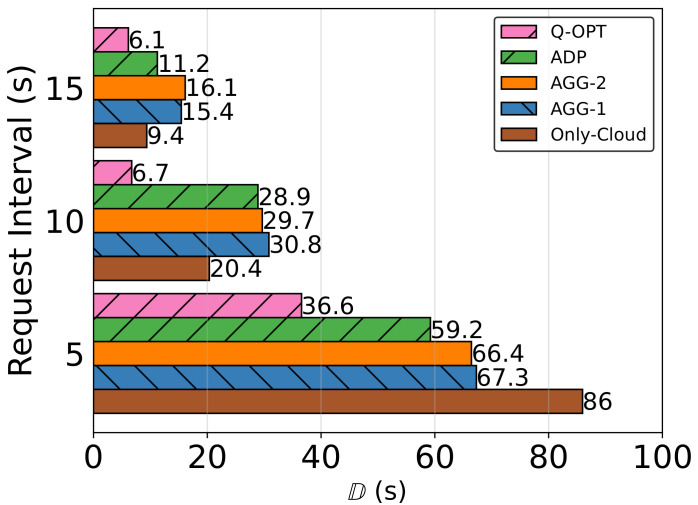
Impact of task inter-arrival time on the objective function D.

**Figure 8 sensors-23-02375-f008:**
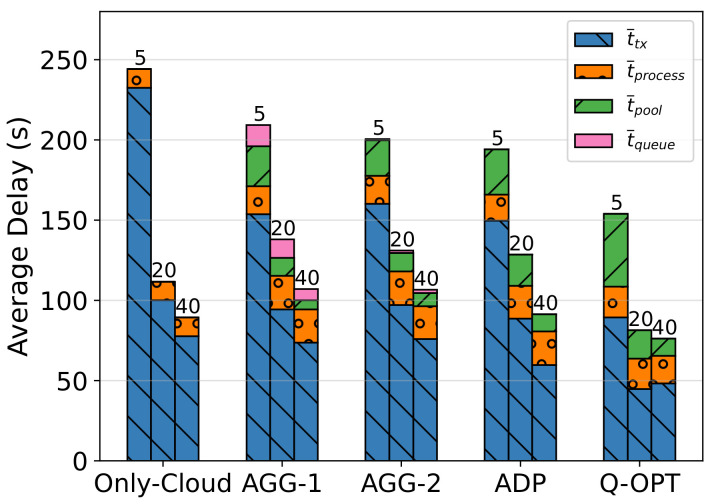
Impact of task inter-arrival time on task delay composition.

**Figure 9 sensors-23-02375-f009:**
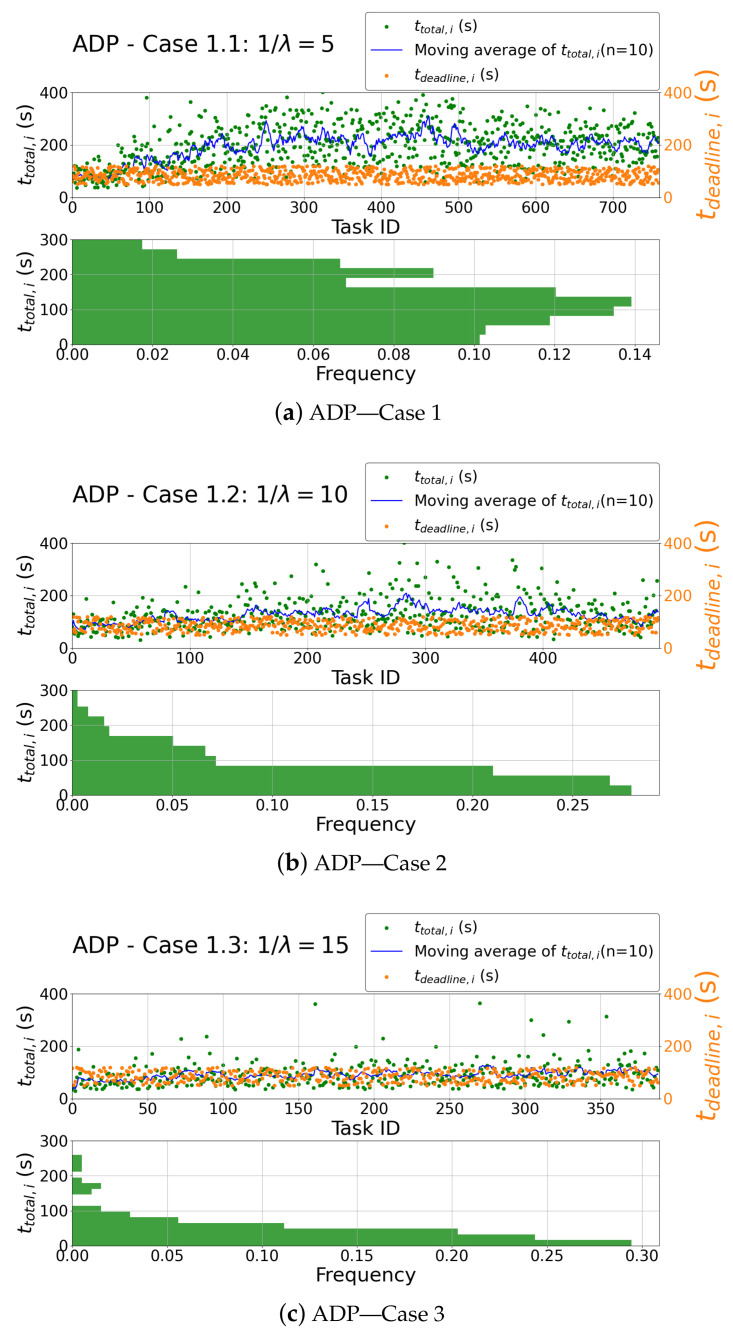
System delay and deadline distribution for ADP.

**Figure 10 sensors-23-02375-f010:**
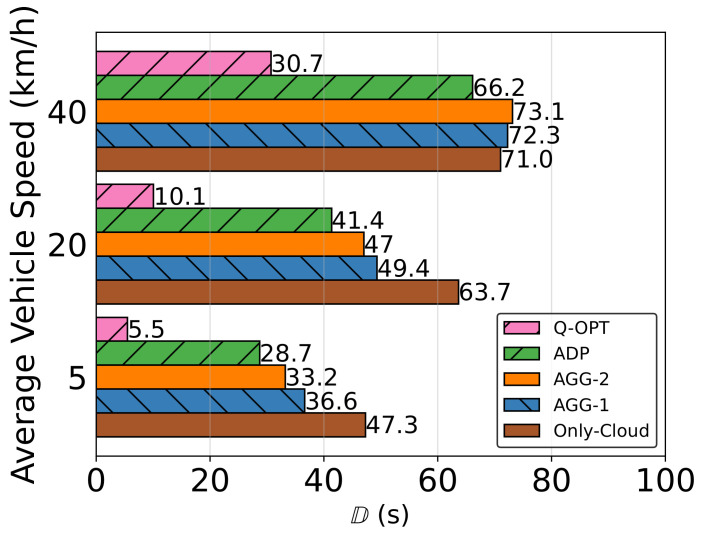
Impact of average vehicle speed on the objective function D.

**Figure 11 sensors-23-02375-f011:**
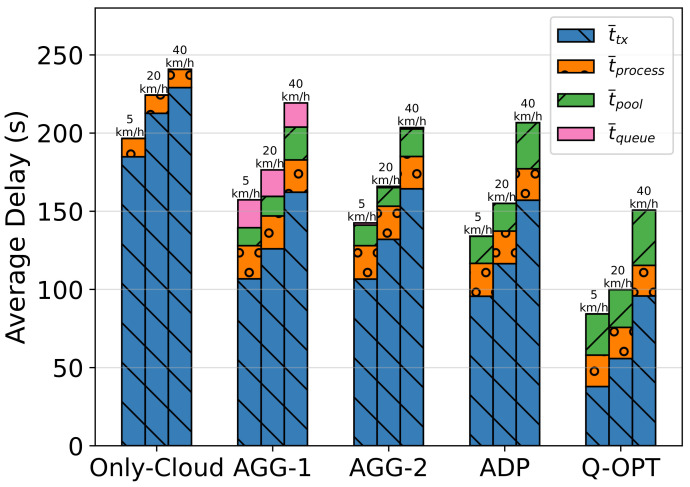
Impact of average vehicle speed on task delay distribution.

**Table 1 sensors-23-02375-t001:** Notations in system modeling.

Symbol	Definition
$U_{i}$	The $i^{th}$ UAV
$V_{i}$	The $i^{th}$ vehicle
$K_{i}$	The $i^{th}$ task
$O_{i,V_{j}}$	If $V_{j}$ is the owner of $K_{i}$, 1; otherwise, 0
$\alpha_{i}$	The priority of $K_{i}$
$D_{i}$	The penalty point of $K_{i}$
$w_{i}$	The weighted penalty point of $K_{i}$
$C_{i}$	If $K_{i}$ is completed, 1; otherwise, 0
$p_{heavy}$	The probability of the task offloaded to the cloud if the pool is heavily occupied
$p_{medium}$	The probability of the task offloaded to the cloud if the pool is moderately occupied
$p_{light}$	The probability of the task offloaded to the cloud if the pool is lightly occupied
$S_{K_{i}}$	The size of $K_{i}$
$P_{i}$	The $i^{th}$ processor
$\beta_{i}$	The processing speed of $P_{i}$
$\beta_{cloud}$	The processing speed of cloud
$S_{Q_{i}}$	The queue size of $P_{i}$
$Q_{i}$	The queue of $P_{i}$
$N_{U}$	The number of UAVs
$N_{K}$	The number of tasks
$N_{P}$	The number of processors
*z*	The extra penalty point if the task deadline is missed

**Table 2 sensors-23-02375-t002:** Time-related notations in problem formulation.

Symbol	Definition
*T*	The number of time slots
$t_{total,i}$	The time elapsed to complete $K_{i}$
$t_{pool,i}$	The time elapsed to be assigned for $K_{i}$
$t_{tx,i}$	The time elapsed to transmit $K_{i}$
$t_{queue,i}$	The time elapsed on processor queue for $K_{i}$
$t_{process,i}$	The time elapsed to process $K_{i}$
$t_{deadline,i}$	The time budget for $K_{i}$ to be processed without any QoS violation, i.e., penalty
$a_{P_{i}}$	The arrival time slot of $P_{i}$
$b_{P_{i}}$	The departure time slot of $P_{i}$
$a_{K_{i}}$	The arrival time slot of $K_{i}$
$c_{K_{i}}$	The completion time slot of $K_{i}$

**Table 3 sensors-23-02375-t003:** Notations in algorithms.

Symbol	Definition
$w_{i,P_{j}}$	$w_{i}$ if $K_{i}$ is assigned to $P_{j}$
$\Theta_{decision}$	The maximum number of candidate solutions that the algorithm will attempt to solve within a time-window
ER	Event for system event received
TP	Controller task pool
TA	Task to be assigned
$List_{D}$	Penalty point list

**Table 4 sensors-23-02375-t004:** Baseline simulation parameters.

**Symbol**	**Value**	**Definition**
λ	0.1 s${}^{-1}$	Exponentially distributed task generation rate
$N_{U}$	9	Number of UAVs
Task size	30–40 MB	Randomly distributed
$\bar{V}$	20 km/h	Average Vehicle Speed
$\beta_{cloud}$	3 MB/s	The processing speed of cloud
$t_{deadline,i}$	50–120 s	The time budget for $K_{i}$ to be processed without any QoS violation (randomly distributed)
$p_{heavy}$	0.9	The probability of the task offloaded to the cloud if the pool is heavily occupied
$p_{medium}$	0.7	The probability of the task offloaded to the cloud if the pool is moderately occupied
$p_{light}$	0.3	The probability of the task offloaded to the cloud if the pool is lightly occupied
$\Theta_{heavy}$	10	The threshold value to classify the controller pool as heavily occupied
$\Theta_{medium}$	7	The threshold value to classify the controller pool as moderately occupied
$\Theta_{light}$	3	The threshold value to classify the controller pool as lightly occupied
$\Theta_{decision}$	10 M	The maximum number of candidate solutions that the algorithm will attempt to solve within a time-window
$P_{tx,U_{i}}$, $P_{tx,V_{i}}$	23 dBm	Transmission power of $U_{i}$, $V_{i}$
$G_{tx,U_{i}}$, $G_{tx,V_{i}}$, $G_{rx,U_{j}}$, $G_{rx,V_{j}}$	3 dBm	Antenna gains
$\sigma^{2}$	−91 dBm	Variance of White Gaussian Noise
*f*	2.4 GHz	The system carrier frequency
ρ	2	Air-to-air path loss exponent
$\eta_{LoS}$, $\eta_{NLoS}$	1 dBm, 20 dBm	The additional LoS,NLoS attenuation factors due to the LoS,NLoS connections
*z*	60	The extra penalty point if the task deadline is missed

**Table 5 sensors-23-02375-t005:** Vehicle parameters.

Role	Class	Proportion	Shape	Color	Priority	$\beta_{i}$	$S_{Q_{i}}$
Owner	A	0.1	Emergency	White	0.7	-	-
Owner	B	0.2	Fire brigade	Red	0.3	-	-
Processor	A	0.06	Trailer	Purple	-	2 MBps	100 MB
Processor	B	0.1	Trailer	Cyan	-	1.5 MBps	70 MB
-	-	0.54	Passenger	Pink	-	-	-

**Table 6 sensors-23-02375-t006:** Impact of task inter-arrival time on QoS and processing characteristics.

	Case ID	1/λ	$\bar{V}$	D	Task Failure Ratio	Cloud Ratio	${\bar{t}}_{pool}$
Only-Cloud	*1.1*	5	20	85.96	0.524	1	0
Only-Cloud	*1.2*	10	20	20.36	0.144	1	0
Only-Cloud	*1.3*	15	20	9.36	0.083	1	0
AGG-1	*1.1*	5	20	67.29	0.396	0.384	24.89
AGG-1	*1.2*	10	20	30.84	0.193	0.024	11.29
AGG-1	*1.3*	15	20	15.41	0.088	0.012	5.71
AGG-2	*1.1*	5	20	66.43	0.405	0.377	22.13
AGG-2	*1.2*	10	20	29.67	0.201	0.029	11.6
AGG-2	*1.3*	15	20	16.12	0.095	0.021	8.22
ADP	*1.1*	5	20	59.22	0.339	0.517	28.14
ADP	*1.2*	10	20	28.9	0.19	0.123	19.57
ADP	*1.3*	15	20	11.23	0.074	0.033	10.67
Q-OPT	*1.1*	5	20	36.56	0.237	0.227	45.46
Q-OPT	*1.2*	10	20	6.71	0.096	0.218	17.65
Q-OPT	*1.3*	15	20	6.1	0.093	0.388	10.72

**Table 7 sensors-23-02375-t007:** Impact of task inter-arrival time on temporal results for task completion.

	Case ID	1/λ	${\bar{t}}_{\mathrm{total}}$	${\bar{t}}_{\mathrm{tx}}$	${\bar{t}}_{\mathrm{queue}}$	${\bar{t}}_{\mathrm{process}}$
Only-Cloud	*2.1*	5	47.31	0.199	1	0
Only-Cloud	*2.2*	20	63.66	0.282	1	0
Only-Cloud	*2.3*	40	71.04	0.358	1	0
AGG-1	*2.1*	5	209.27	153.71	13	17.46
AGG-1	*2.2*	10	138.02	94.43	12	20.8
AGG-1	*2.3*	15	107.07	73.59	7	20.76
AGG-2	*2.1*	5	200.54	160.24	1	17.45
AGG-2	*2.2*	10	131.07	97.19	2	20.77
AGG-2	*2.3*	15	106.64	75.82	2	20.56
ADP	*2.1*	5	194.17	149.76	0	16.27
ADP	*2.2*	10	128.57	88.57	0	20.4
ADP	*2.3*	15	91.34	59.75	0	20.87
Q-OPT	*2.1*	5	94.96	33.96	0	20.73
Q-OPT	*2.2*	10	73.05	35.7	0	20.35
Q-OPT	*2.3*	15	70.44	39.34	0	19.06

**Table 8 sensors-23-02375-t008:** Impact of average vehicle speed on QoS and processing characteristics.

	Case #	1/λ	$\bar{V}$	D	Task Failure Ratio	Cloud Ratio	${\bar{t}}_{\mathrm{pool}}$
Only-Cloud	*2.1*	8	5	47.31	0.199	1	0
Only-Cloud	*2.2*	8	20	63.66	0.282	1	0
Only-Cloud	*2.3*	8	40	71.04	0.358	1	0
AGG-1	*2.1*	8	5	36.63	0.153	0.042	11.54
AGG-1	*2.2*	8	20	49.35	0.211	0.01	12.45
AGG-1	*2.3*	8	40	72.29	0.42	0.117	21.02
AGG-2	*2.1*	8	5	33.22	0.146	0.03	13.11
AGG-2	*2.2*	8	20	47.03	0.212	0.015	11.54
AGG-2	*2.3*	8	40	73.14	0.478	0.121	17.67
ADP	*2.1*	8	5	28.71	0.137	0.107	17.31
ADP	*2.2*	8	20	41.35	0.197	0.086	17.64
ADP	*2.3*	8	40	66.15	0.399	0.256	29.53
Q-OPT	*2.1*	8	5	5.51	0.055	0.146	26.27
Q-OPT	*2.2*	8	20	10.08	0.033	0.133	24.11
Q-OPT	*2.3*	8	40	30.69	0.143	0.188	35.27

## Data Availability

https://github.com/onurklngc/drone-network-onos-and-mininet/tree/master.

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
