# Peer review of "Prioritization Based Task Offloading in UAV-Assisted Edge Networks†"

_sensors, 2023, doi:10.3390/s23052375_

Round 1
Reviewer 1 Report
The topic is interesting and relevant to the field. The problem is important and addressed with higher degree of novelty. The proposed methods are well-explained. The experiments are convincing and validated numerically. The scientific knowledge of the authors is sound and technical writing is adequate. But some minor suggestions are given below to address.
· Abstract needs improvements, as it does not indicate the improvements in your proposal.
· The caption of Table 1 should be revised. Like “Notations used in System Modeling”. Similar comment for table 2 and 3.
· The system model is illustrated by a sequence diagram in fig 2 that is too generic. Its quality should be improved by replacing it with new one that shows the standard Task offloading scenarios.
· Some equations (3-5, 11-14, and 17-21) needs more and individual explanation.
· The algorithms in fig 3 and 4 should be contained in a table. Moreover, they seem open ended. The statements in Algorithm (fig 3) need to be concise.
· There are more than 20 equations in the manuscript. It is suggested to keep the formulas explaining the main idea behind system modelling and move the rest of the equations to an Appendix section as the end.
· In fig 7 (a) an (b) should be separated to enhance the visibility.
· The subheadings in section 7.3.1 should be numbered alphabetically or in Romans.
· The limitations of the proposed technique should be discussed in conclusion.
· Again, proofread the article for English errors.
· To increase the research impact, enhance the quality of references. Some recent works on task offloading in Edge Networks are given below to add.
LiMPO: lightweight mobility prediction and offloading framework using machine learning for mobile edge computing.
Author Response
· Abstract needs improvements, as it does not indicate the improvements in your proposal.
We thank the reviewer for pointing out this important missing element. Related improvements from our proposal are now added to the end of the abstract.
· The caption of Table 1 should be revised. Like “Notations used in System Modeling”. Similar comment for table 2 and 3.
All captions including Table 1-3, A11 are checked, revised and improved.
· The system model is illustrated by a sequence diagram in fig 2 that is too generic. Its quality should be improved by replacing it with new one that shows the standard Task offloading scenarios.
More details with notations, annotations and decision steps are added to improve this figure.
· Some equations (3-5, 11-14, and 17-21) needs more and individual explanation.
We have improved the already-existing explanations and also added one for the equations lacking an explanation.
· The algorithms in fig 3 and 4 should be contained in a table. Moreover, they seem open ended. The statements in Algorithm (fig 3) need to be concise.
Those floating elements are now placed into Table (Table 4-5). We also improved the notations to make the statements more consise. We thank the reviewer again, to point out the error in the algorithm statement. Now, end statements are added for avoiding open-ended algorithm execution.
· There are more than 20 equations in the manuscript. It is suggested to keep the formulas explaining the main idea behind system modelling and move the rest of the equations to an Appendix section as the end.
The equations related to the channel models are not central to the main contributions of the paper as correctly identified by the reviewer. Therefore, we moved those equations to the Appendix.
· In fig 7 (a) an (b) should be separated to enhance the visibility.
They are now separated into Fig 5 and Fig 6.
· The subheadings in section 7.3.1 should be numbered alphabetically or in Romans.
As suggested by the reviewer, Roman numbers are added to improve formatting.
· The limitations of the proposed technique should be discussed in conclusion.
The limitations based on the algoritm are added and now discussed in the Conclusion section.
· Again, proofread the article for English errors.
We have proofread our work multiple times and fixed the grammatical errors and typos.
· To increase the research impact, enhance the quality of references. Some recent works on task offloading in Edge Networks are given below to add.
The suggested paper as well as two others is added to the related work.
- LiMPO: lightweight mobility prediction and offloading framework using machine learning for mobile edge computing.
- Sacco, A.; Esposito, F.; Marchetto, G.; Montuschi, P. A Self-Learning Strategy for Task Offloading in UAV Networks. IEEE Transactions on Vehicular Technology 2022, 71, 4301–4311. https://doi.org/10.1109/TVT.2022.3144654
- Kang, H.; Chang, X.; Miši ́c, J.; Miši ́c, V.B.; Fan, J.; Liu, Y. Cooperative UAV Resource Allocation and Task Offloading in Hierarchical Aerial Computing Systems: A MAPPO Based Approach. IEEE Internet of Things Journal 2023, pp. 1–1. https://doi.org/10.1109/JIOT.2023.3240173
Reviewer 2 Report
This is a well-written Prioritization Based Task Offloading in UAV-Assisted Edge Networks manuscript. However, some improvements are needed.
1. Please highlight the three heuristic algorithms used for the abstract.
2. Some references are absolute (2005, 2010, 2014,2015). Please look for the most recent references.
3. Table 7, please include a column for the speed.
4. As for the conclusion, please suggest the recommended heuristic algorithm for the selected parameter.
Author Response
1. Please highlight the three heuristic algorithms used for the abstract.
We have added text to highlight the implementation of three algoritms into the abstract.
2. Some references are absolute (2005, 2010, 2014,2015). Please look for the most recent references.
2005 -> This one is for the Wifi standard.
2010 -> Referencing Mininet.
The following three papers published in recent years are added:
2022 -> Zaman, S.K.; Jehangiri, A.; Maqsood, T.; Shuja, J.; Ahmad, Z.; Umar, A. LiMPO: lightweight mobility prediction and offloading framework using machine learning for mobile edge computing. Cluster Computing 2022. https://doi.org/10.1007/s10586-021-03518-7
2022 -> Sacco, A.; Esposito, F.; Marchetto, G.; Montuschi, P. A Self-Learning Strategy for Task Offloading in UAV Networks. IEEE Transactions on Vehicular Technology 2022, 71, 4301–4311. https://doi.org/10.1109/TVT.2022.3144654
2023 -> Kang, H.; Chang, X.; Miši ́c, J.; Miši ́c, V.B.; Fan, J.; Liu, Y. Cooperative UAV Resource Allocation and Task Offloading in Hierarchical Aerial Computing Systems: A MAPPO Based Approach. IEEE Internet of Things Journal 2023, pp. 1–1. https://doi.org/10.1109/JIOT.2023.3240173
3. Table 7, please include a column for the speed.
The speed is now added to Table 8 and the request rate is added to Table 10 for consistency.
4. As for the conclusion, please suggest the recommended heuristic algorithm for the selected parameter.
A comparison of algorithms are now added to the conclusion.